# Elucidating the interplay between metabolites and microorganisms in the spermosphere of common bean (*Phaseolus vulgaris* L.) seeds

Chandrodhay Saccaram,[1] Marie Simonin,[2] Stéphanie Boutet,[1] Céline Brosse,[1] Shuang Peng,[1] Tracy François,[1] Boris Collet,[1] François Perreau,[1] Delphine Sourdeval,[1] Coralie Marais,[2] Matthieu Barret,[2] Loïc Rajjou,[1] Massimiliano Corso[1]

**ABSTRACT**  The spermosphere, the dynamic interface surrounding germinating seeds, is shaped by the intricate interplay between seed-exuded natural compounds and seed-associated microbial communities. In this work, we provide the first comprehensive metabolomic and microbiome characterization of common bean (*Phaseolus vulgaris*) spermosphere of eight genotypes produced in two contrasted production regions. Through an integrated approach, we explored the metabolomic and microbiota composition in the spermosphere of germinating common bean seeds and elucidated their environmental and genotype regulation. We detected and analyzed 2,467 metabolite features (Mf) through untargeted metabolomics categorized into fourteen metabolic categories, highlighting the prevalence of amino acids, flavonoids, and terpenoids. Genotype was the key factor influencing the chemical composition of the spermosphere. Furthermore, we identified 19 bacterial families and 23 fungal families inhabiting the spermosphere, with both genotype and seed production location exerting varying degrees of influence on microbial community composition. Through a multiscale integrated approach, we revealed specific associations between metabolites and microorganisms, such as negative correlation between flavonoids and *Bacillus* spp., emphasizing the genotype-dependent nature of these interactions. This comprehensive investigation sheds light on the mechanisms underlying seed germination and the complex interactions between plant genotypes, seed exudates, environmental conditions, and microbial communities in the spermosphere. These findings provide a framework for developing innovative strategies to promote seed health and sustainable crop production.

**IMPORTANCE**  The spermosphere, the dynamic interface around germinating seeds, is shaped by the intricate interplay between seed-exuded compounds and microbial communities. Despite the importance of these interactions for eventual seedling emergence and health, little knowledge is available on the subject. We are the first to comprehensively analyze the chemical and microbial diversity of the spermosphere of *Phaseolus vulgaris* (common bean). We identified thousands of primary and specialized metabolites, highlighting their diversity but largely unknown roles in germinating seed-environment interactions. We revealed significant genotype-driven differences in the chemical composition as well as the influence of both genotype and seed production location on microbial community structure in the spermosphere. Our metabolome-microbiome integrative analysis suggests that common bean shapes the spermosphere microbiome through specific seed exudates. This research advances our understanding of the metabolic capabilities and ecological roles of seed microbiota within the spermosphere, contributing to our understanding of seed health and vigor.

**Peer Reviewers** Malak Tfaily, The University of Arizona, Tucson, Arizona, USA; Toshihiro Obata, University of Nebraska-Lincoln, Lincoln, Nebraska, USA

Address correspondence to Loïc Rajjou, loic.rajjou@agroparistech.fr, or Massimiliano Corso, massimiliano.corso@inrae.fr.

The authors declare no conflict of interest.

See the funding table on p. 17.

KEYWORDS   seeds, spermosphere, germination, metabolome, microbiome, *Phaseolus vulgaris*

Seeds play a crucial role in plant propagation and agricultural sustainability, with their quality and vigor significantly impacting crop yield and resilience in diverse environmental conditions (1, 2). During germination, seeds release complex mixtures of organic and inorganic compounds into their surrounding environment, creating a localized microenvironment known as the spermosphere, i.e., a zone of increased microbial activity around germinating seeds (3–6). While extensive research has focused on plant microbiomes in the phyllosphere and rhizosphere, the spermosphere and its microbial and chemical composition have received limited attention (3, 5, 6). Seeds serve as vital conduits for transmitting microorganisms between plant generations, influencing the establishment of the plant microbiome (7). Previous studies have shown that seed microbial communities are influenced by seed genotype and production environment (8, 9), possibly due to the diversity of primary and specialized metabolites accumulated in seeds (10). Seed exudates during germination further contribute to shaping the spermosphere microbiome, similar to plant-microorganism interactions in the rhizosphere mediated by root exudates (11), but this remains to be investigated. For instance, the isoflavone daidzein influenced microbial community structure in the rhizosphere (12, 13). Furthermore, the flavanone naringenin has induced chemotaxis in rhizobacteria *Aeromonas* sp (14).

Despite the significance of metabolic transition during seed germination (15), limited information exists on the chemical (16) and microbial composition of the seed spermosphere (17), which are likely to play essential roles in seed interactions with the environment. Seed germination and seedling emergence have been shown to favor microbial taxa with fast-growing abilities (18), suggesting the importance of elucidating interactions in the spermosphere.

Common bean (*Phaseolus vulgaris* L.) is a globally consumed grain legume rich in nutraceutical metabolites and proteins (19, 20). The domestication of common bean led to phenotypic changes affecting both the seed microbiome community structure and chemical composition (21, 22). Studying common bean interaction with the environment is crucial to improve seed vigor. In this context, seeds of this species provide a suitable model to explore the interactions between metabolites and microorganisms in the spermosphere (23).

In this work, we designed a simplified experimental setup to collect the spermosphere upon seed imbibition for studying metabolome and microbiome interplay. We hypothesized that seeds play a central role in shaping the spermosphere by influencing both exudate composition and associated microbial communities. To investigate this, we (i) characterized the chemical and microbial composition of the common bean spermosphere, (ii) identified the factors (namely genotype and seed production environment) that influence spermosphere composition and microbial assembly, and (iii) explored associations between seed-borne microorganisms and seed-exuded metabolites. Using integrated metabolomic and microbiome multi-omics data, we characterized, for the first time, the metabolic and microbial composition of the common bean spermosphere, as well as the complex interactions between metabolites and microorganisms. We demonstrated that the chemical composition of the common bean seed spermosphere is primarily influenced by genotype, while seed-borne microbial communities are shaped by both genotype and the seed production environment. Integration of multi-omics data enabled the identification of potential associations between spermosphere metabolites and both beneficial and pathogenic microorganisms. These findings provide new insights into the ecological roles of spermosphere metabolites and microbiota in common bean, enhancing our understanding of seed health and vigor.

## RESULTS

### Chemical and microbial diversity in the common bean spermosphere

#### Chemical diversity in common bean spermosphere

Metabolite diversity of the germinating seed spermosphere of eight common bean genotypes produced in two locations (24, 25) was analyzed using untargeted metabolomics (LC–MS/MS) in electrospray ionization in positive (ESI+) and negative (ESI−) modes. The metabolomic analyses allowed the detection of 2,467 ions, or metabolite features (Mf), with a putative or unknown annotation. LC-MS/MS data were used to build a molecular network (Fig. 1A; see Fig. S4 at https://doi.org/10.6084/m9.figshare.26117287.v1) (26, 27) to improve the assignment of each Mf to a metabolic category (Fig. 1B; see Table S1 at https://doi.org/10.6084/m9.figshare.26117287.v1). This analysis allowed the assignment of a total of 506 Mf (21% of the spermosphere metabolome) to a known metabolic category. Mf identified in the germinating common bean seed spermosphere have been assigned to 14 putative major metabolic categories (Fig. 1B) and specific classes (see Table S1 at https://doi.org/10.6084/m9.figshare.26117287.v1). The categories in which the highest number of Mf were detected correspond to amino acid (aa) and derivatives (140), flavonoids (126), and terpenoids and derivatives (55). Amino acids and derivatives were mainly represented by very small peptides (102), flavonoids were mainly represented by flavonols (44) and flavan-3-ols (30), and jasmonic acid and derivatives (7) were predominant in the plant hormone category (see Table S1 at https://doi.org/10.6084/m9.figshare.26117287.v1).

#### Spermosphere microbial assemblages were represented by 19 bacterial and 23 fungal families

Spermosphere bacterial and fungal communities were studied, using the exact same samples employed for metabolomic analysis, by a metabarcoding community profiling approach (see Table S2 and S3 at https://doi.org/10.6084/m9.figshare.26117287.v1).

The bacterial community in the spermosphere of common bean samples primarily comprised two prominent phyla: *Pseudomonadota* and *Bacillota*, with average relative abundances of 75.8% and 23.2%, respectively, across all samples (Fig. 2A; see Table S4 at https://doi.org/10.6084/m9.figshare.26117287.v1). Nineteen bacterial families were identified, with *Erwiniaceae* being notably substantial with average relative abundance of 55.0% across all samples (Fig. 2B; see Table S4 at https://doi.org/10.6084/m9.figshare.26117287.v1 ). Within this family, *Pantoea* with average relative abundance of 33.7% across all samples emerged as the most abundant genus (Fig. 2C; see Table S4 at https://doi.org/10.6084/m9.figshare.26117287.v1) with *Pantoea agglomerans* (B_ASV2) showing high conservation across samples, being present in over 70% of all samples. This highlights its significance as a core component of the spermosphere taxa.

*Ascomycota* and *Basidiomycota,* with an average relative abundance across all samples of 97.1% and 2.9%, respectively, were two fungal phyla in the common bean spermosphere (Fig. 2E; see Table S5 at https://doi.org/10.6084/m9.figshare.26117287.v1). Fungi were further categorized into 23 distinct families, with the *Pleosporaceae* family being notably overrepresented with average relative abundance of 64.3% across all samples of germinating bean spermosphere (Fig. 2F; see Table S5 at https://doi.org/10.6084/m9.figshare.26117287.v1). *Alternaria* and *Stemphylium* (*Pleosporaceae* family) emerged as the most abundant fungal genera across the common bean spermosphere, boasting average relative abundances of 34.0% and 29.9% across all samples, respectively (Fig. 2G; see Table S5 at https://doi.org/10.6084/m9.figshare.26117287.v1). Several fungal species were present in over 70% of the common bean genotype samples, thereby forming the core fungal taxa within the spermosphere (see Table S5 at https://doi.org/10.6084/m9.figshare.26117287.v1). These included non-pathogenic and potential pathogenic filamentous fungi, such as *Gibberella intricans*, *Stemphylium herbarum*, *Botrytis fabae*, *Alternaria infectoria*, an *Alternaria sp., Cladosporium delicatulum*,

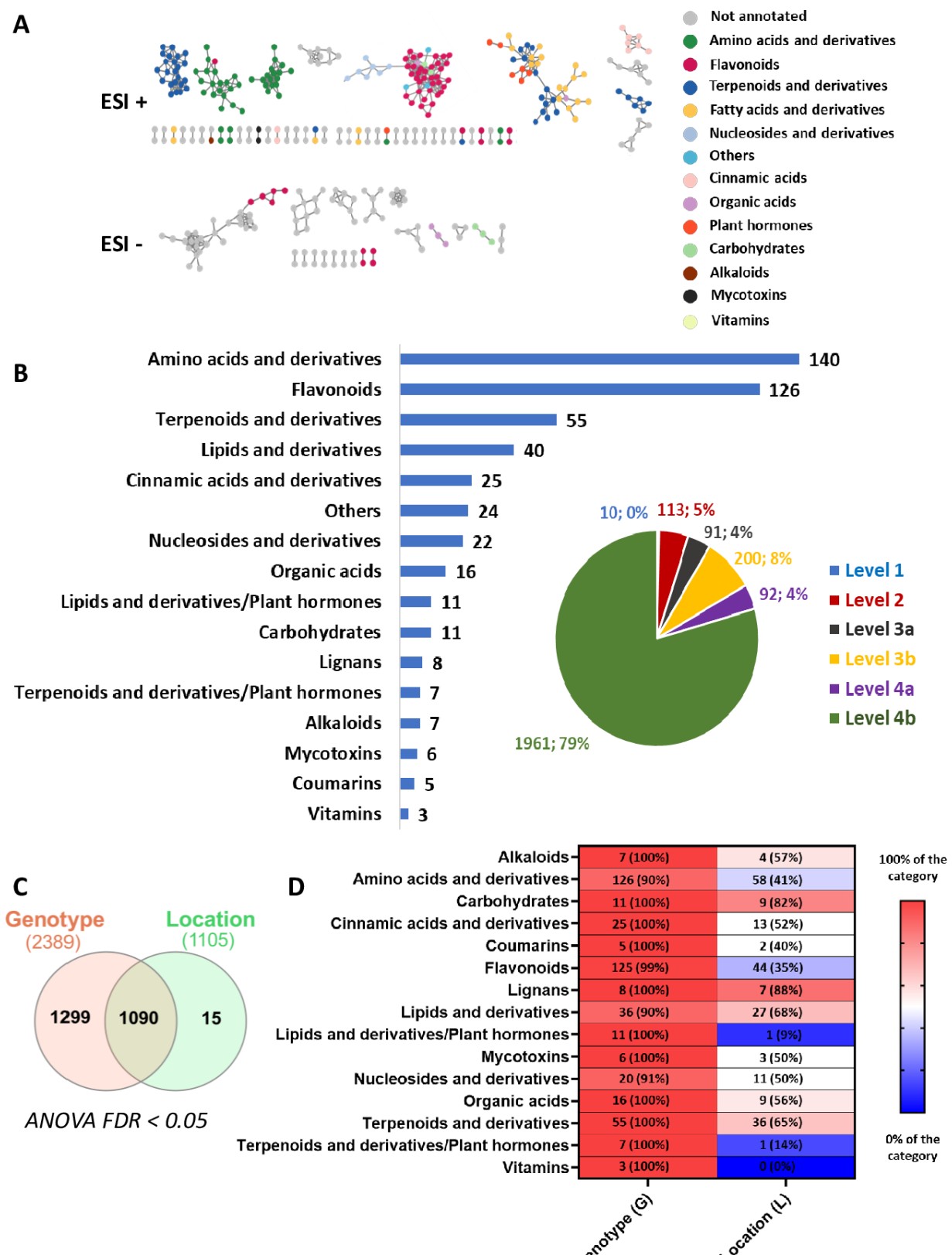

**FIG 1** Chemical diversity in the spermosphere of germinating seeds from eight genotypes of common beans produced in two different locations. Analysis was performed on a data set of 48 samples from eight common bean genotypes grown in two locations (Gers and Maine et Loire). (A) A subset of the molecular networks is shown for metabolite analysis carried out in positive (ESI+) and negative (ESI−) ionization modes. Different colors correspond to distinct metabolic

Fig 1 (Continued)

categories. Metabolites are grouped based on their chemical structures. Cosine similarity scores of 0.8 and 0.7 were used for ESI + and ESI−, respectively. *Plant hormones contain two categories: lipid-derived hormones and terpenoid-derived hormones. For representation purposes, they were given the same colors in the molecular network. (B) Histogram showing the annotated/known metabolic categories identified in common bean seeds. The number of metabolites that belong to each category is indicated. Percentage of annotated metabolites by confidence levels (levels 1–4) is also represented in the sector diagram. Metabolites were annotated by comparing with GNPS public spectral libraries as well as internal homemade libraries. Further putative annotations were carried out by molecular networks. (C) Venn diagram of differential metabolite features (FDR adjusted $P < 0.05$) identified by two-way ANOVA on log10 transformed data set containing 2,467 metabolic features across 48 samples. (D) Annotated differential metabolites identified by two-way ANOVA (FDR adjusted $P < 0.05$) on log10 transformed data set.

as well as two *Cladosporium* spp. (F_ASV8 and F_ASV105), and Basidiomycota yeasts, such as *Vishniacozyma victoriae*.

### Contrasted patterns in alpha diversity for bacteria and fungi

Bacterial and fungal richness levels were then assessed. Bacterial richness, determined through *gyrB* sequence analysis, unveiled a median of 25 ASVs, with a large variability spanning from 2 to 46 ASVs among samples (Fig. 2D; see Table S6 at https://doi.org/10.6084/m9.figshare.26117287.v1). Likewise, fungal richness exhibited a median richness of 29 ASVs, encompassing a range of 3 to 43 ASVs (Fig. 2H; see Table S7 at https://doi.org/10.6084/m9.figshare.26117287.v1).

Bacterial richness was mainly influenced by the seed production location (Fig. 2D; ANOVA, $P < 0.01$), whereas fungal richness was significantly affected by location, genotype, and their interaction (Fig. 2H; ANOVA, $P < 0.05$). Overall, both bacterial and fungal richness were higher in seeds produced in Maine-et-Loire compared with those from the Gers region (Fig. 2D and H).

### The metabolic landscape and microbial community structure of the germinating common bean spermosphere are influenced by both the genotype and the site of production

Statistical analyses, including ANOVA ($P < 0.05$; Fig. 1I and J), PCA, and PERMANOVA (Fig. 3A and G), revealed that genotype is the primary factor influencing spermosphere

**TABLE 1** Network analysis of pairwise metabolite-microorganism correlation network showing top 15 nodes by degree[a]

| Node | Group | Species/Metabolites | Genus/Category | Degree |
|------|-------|---------------------|----------------|--------|
| B_ASV99 | Bacteria | *Pantoea ananatis* | *Pantoea* | 428 |
| B_ASV19 | Bacteria | *Bacillus megaterium* | *Bacillus* | 397 |
| B_ASV308 | Bacteria | Unclassified Actinobacteria | Unclassified *Actinobacteria* | 308 |
| B_ASV162 | Bacteria | Unclassified Firmicute | Unclassified Firmicute | 285 |
| B_ASV27 | Bacteria | *Bacillus megaterium* | *Bacillus* | 263 |
| B_ASV132 | Bacteria | *Paenibacillus sp.* | *Paenibacillus* | 251 |
| B_ASV10 | Bacteria | *Bacillus sp.* | *Bacillus* | 240 |
| B_ASV91 | Bacteria | *Pantoea sp.* | *Pantoea* | 222 |
| B_ASV14 | Bacteria | *Bacillus megaterium* | *Bacillus* | 217 |
| B_ASV40 | Bacteria | *Bacillus toyonensis* | *Bacillus* | 215 |
| F_ASV3 | Fungi | *Botrytis fabae* | *Botrytis* | 156 |
| F_ASV2 | Fungi | *Stemphylium herbarum* | *Stemphylium* | 155 |
| F_ASV27 | Fungi | *Gibellulopsis nigrescens* | *Gibellulopsis* | 129 |
| F_ASV13 | Fungi | Unclassified Dothideomycetes | Unclassified Dothideomycetes | 124 |
| F_ASV9 | Fungi | *Alternaria infectoria* | *Alternaria* | 121 |

[a]Node: Connection point of the network. it can be either the unique ID of a metabolite or microbial species. Group: Specifying the type of variable represented by the node (bacteria, fungi or metabolites). Species/Metabolites: Identity at the level of species for microbial nodes and identity obtained through comparison with metabolic libraries for metabolite nodes. Genus/Category: Identity at the level of genus for microbial nodes and metabolic categories for metabolite nodes. Degree: Number of connections or edges the node has to other nodes.

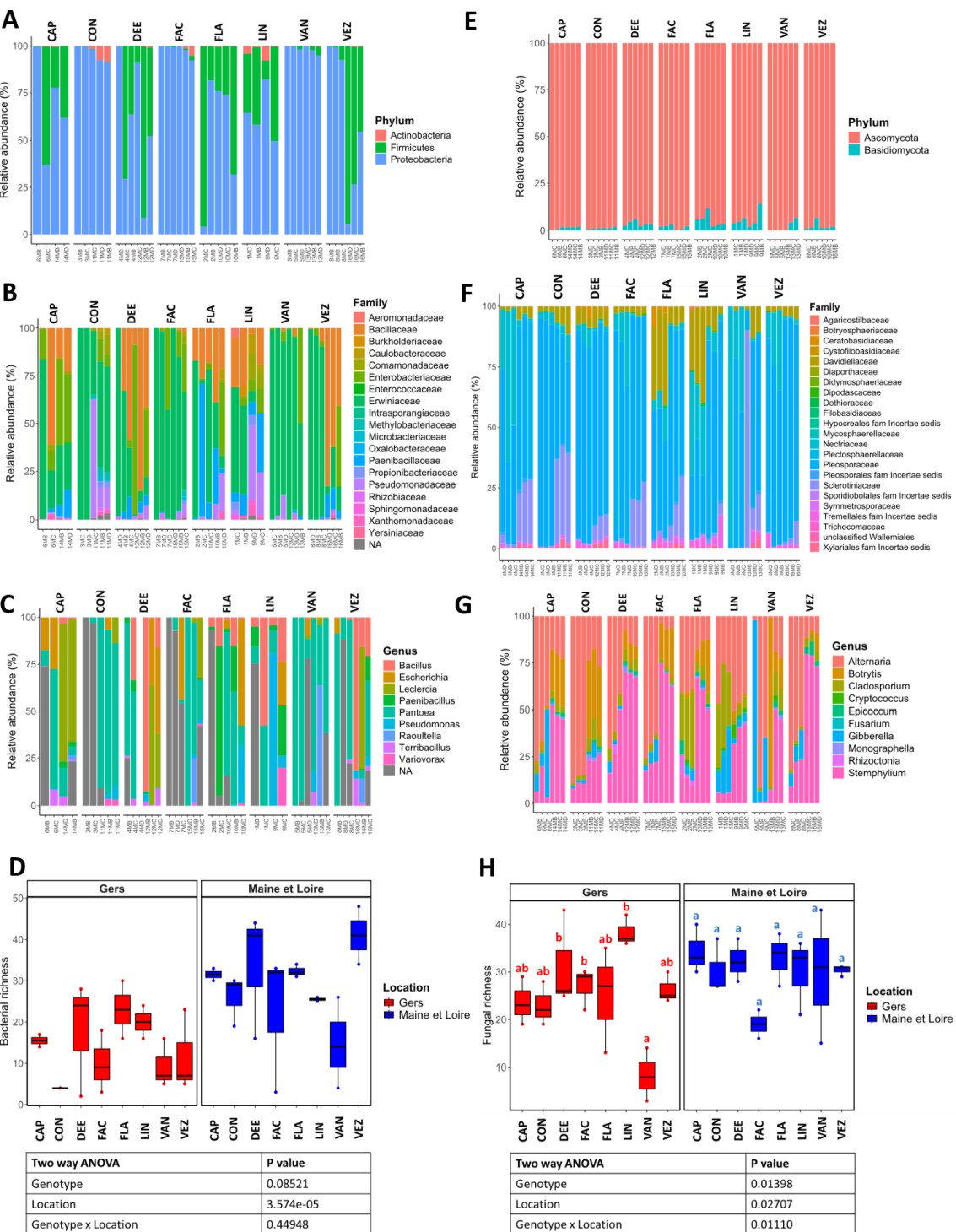

**FIG 2** Microbial diversity in the spermosphere of germinating seeds from eight genotypes of common beans produced in two different locations. Analyses were conducted on rarefied data sets: *gyrB* (bacteria) at 1,250 reads per sample (resulting in 42 samples) and ITS (fungi) at 5,000 reads per sample (retaining all 48 samples). (A) and (E) Relative abundance of bacterial and fungal phyla, respectively. (B) and (F) Relative abundance of bacterial and fungal families, respectively. (C) and (G) Relative abundance of 10 most represented bacterial and fungal genera, respectively. (D) and (H) Measure of bacterial and fungal α-diversity through observed richness (number of predicted species) produced in Gers (red box plots) and Maine & Loire (blue box plots). Box plots display the data distribution through their quartiles, and whiskers are used to indicate variability outside the upper and lower quartiles. The median marks the midpoint of the data and is indicated by the black line.

metabolic diversity across all metabolic categories, whereas location had a comparatively limited effect on the metabolite landscape. Heatmap of relative abundance of metabolites (Fig. 3A) and metabolic classes (see Fig. S5 at https://doi.org/10.6084/m9.figshare.26117287.v1) in the spermosphere further confirmed separation of samples mainly according to genotypes.

Microbiome analyses of the common bean spermosphere revealed a comparable impact of genotype on bacterial (23%; Fig. 3C, D and G) and fungal (19%; Fig. 3E, F and G) community diversity. In contrast, location had a stronger effect on fungal diversity, accounting for 43% of the variation (Fig. 3E and G). While hierarchical clustering based on bacterial abundance showed no clear pattern in distribution of samples (Fig. 3D), fungal abundance data showed a clear distribution of samples between the two locations (Fig. 3F).

## Metabolome-microbiome integration analysis

### Multiscale network analysis

The relationship between metabolome and microbiome in common bean spermosphere was explored through a correlation network analysis approach (28). Pairwise Spearman metabolite-microorganism and microorganism-microorganism correlations ($r \geq |0.5|$; $P < 0.05$) were selected to create a network (Fig. 4A; see Table S10 at https://doi.org/10.6084/m9.figshare.26117287.v1). The resulting network encompassed a total of 1,736 nodes, comprising metabolites (1,559), bacteria (107), and fungi (70), connected by 6,225 edges representing the correlations (Fig. 4A and B). Negative correlation coefficients ranged from −0.825 to −0.5, while the positive coefficients spanned from 0.5 to 0.909.

**TABLE 2** Enrichment analysis of microbial genus overrepresented in correlation subnetwork of different metabolic categories as compared with spermosphere[a]

| Microbial genus | Target size | Target occurrence | Target frequency | Reference size | Reference occurrence | Reference frequency | Fold enrichment | P-Value |
|---|---|---|---|---|---|---|---|---|
| Enriched in amino acids and derivatives subnetwork | | | | | | | | |
| Bacillus | 39 | 13 | 33.3% | 118 | 24 | 20.3% | 1.6 | 0.0146 |
| Variovorax | 39 | 3 | 7.7% | 118 | 3 | 2.5% | 3.0 | 0.0342 |
| Alternaria | 23 | 4 | 17.4% | 82 | 6 | 7.3% | 2.4 | 0.0492 |
| Enriched in carbohydrates subnetwork | | | | | | | | |
| Bacillus | 24 | 10 | 41.7% | 118 | 24 | 20.3% | 2.0 | 0.0062 |
| Enriched in cinnamic acids and derivatives subnetwork | | | | | | | | |
| Cladosporium | 15 | 3 | 20.0% | 82 | 5 | 6.1% | 3.3 | 0.0403 |
| Enriched in coumarins subnetwork | | | | | | | | |
| Cladosporium | 11 | 4 | 36.4% | 82 | 5 | 6.1% | 6.0 | 0.0009 |
| Gibberella | 11 | 2 | 18.2% | 82 | 3 | 3.7% | 5.0 | 0.0460 |
| Enriched in flavonoids subnetwork | | | | | | | | |
| Bacillus | 34 | 13 | 38.2% | 118 | 24 | 20.3% | 1.9 | 0.0031 |
| Cladosporium | 24 | 4 | 16.7% | 82 | 5 | 6.1% | 2.7 | 0.0241 |
| Enriched in lignans subnetwork | | | | | | | | |
| Pantoea | 6 | 3 | 50.0% | 118 | 9 | 7.6% | 6.6 | 0.0056 |
| Enriched in lipids and derivatives subnetwork | | | | | | | | |
| Variovorax | 9 | 3 | 33.3% | 118 | 3 | 2.5% | 13.1 | 0.0150 |
| Enriched in organic acid subnetwork | | | | | | | | |
| Pantoea | 8 | 3 | 37.5% | 118 | 9 | 7.6% | 4.9 | 0.0144 |
| Enriched in terpenoids and derivatives subnetwork | | | | | | | | |
| Bacillus | 29 | 10 | 34.5% | 118 | 24 | 20.3% | 1.7 | 0.0312 |
| Cladosporium | 15 | 3 | 20.0% | 82 | 5 | 6.1% | 3.3 | 0.0403 |

[a]Microbial genus: Enriched microbial genus in specific metabolic category correlation subnetwork. Target size: Number of species belonging to specific microbial kingdom present in correlation metabolic subnetwork (Target size >5 were retained). Target occurrence: Number of species belonging to overrepresented microbial genus in correlation metabolic subnetwork. Target frequency: Frequency of overrepresented genus in correlation metabolic subnetwork. Reference size: Total number of species belonging to specific microbial kingdom present in spermosphere. Reference occurrence: Number of species belonging to overrepresented microbial genus in spermosphere. Reference frequency: Frequency of overrepresented genus in spermosphere. Fold enrichment: Enrichment score (score >1.5 were retained). P-value: Statistical significance of microbial enrichment test (hypergeometric test).

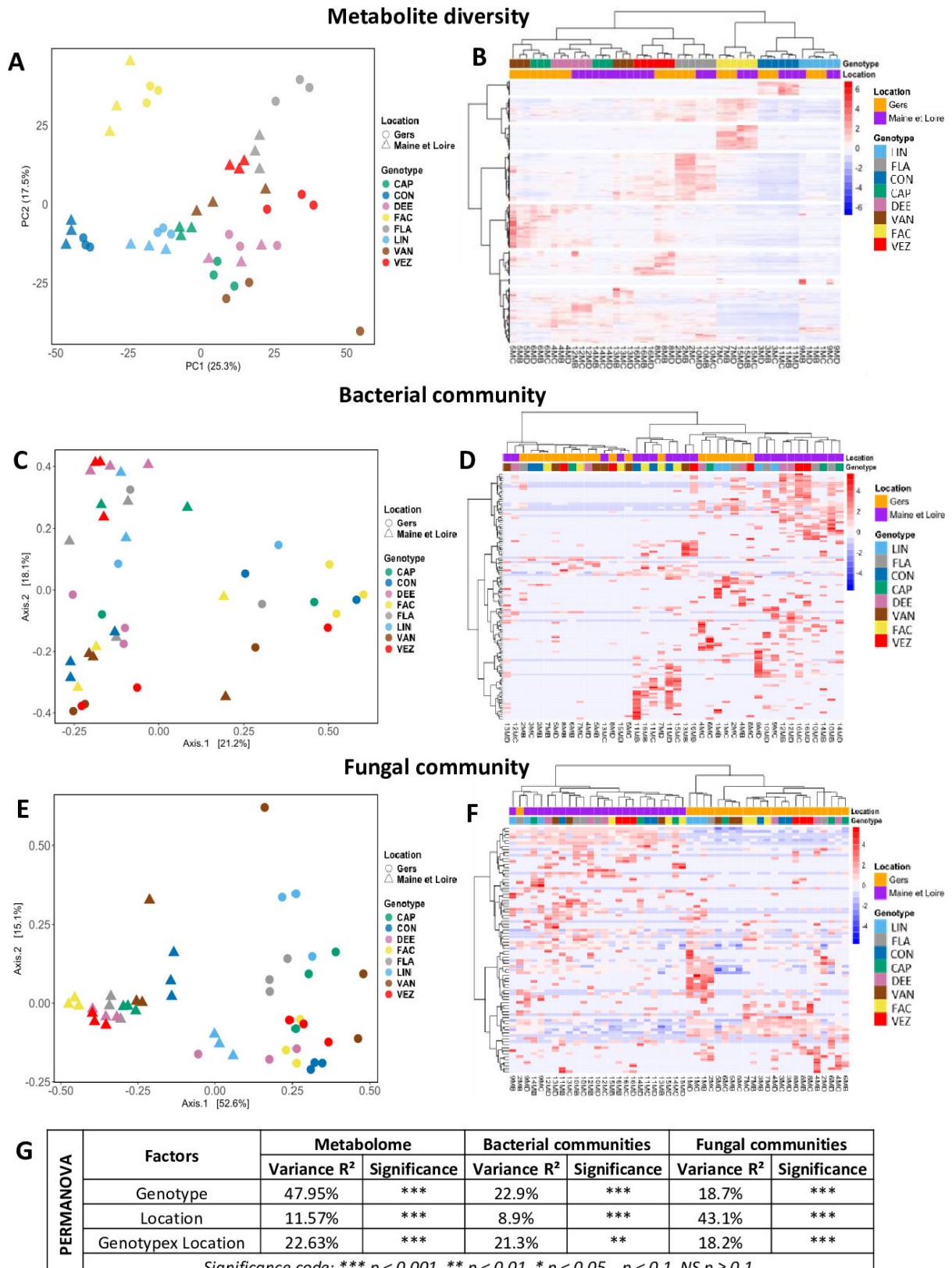

**FIG 3** Factors influencing microbial structure and metabolomic landscape in common bean spermosphere. Analyses were conducted on rarefied data sets: *gyrB* at 1,250 reads per sample (resulting in 42 samples) and ITS at 5,000 reads per sample (retaining all 48 samples), alongside metabolomic data from 48 samples. (A) Principal component analysis (PCA) scores plot of original normalized data of spermosphere metabolome distribution of eight common bean seed genotypes (Continued on next page)

**Fig 3 (Continued)**

(represented by different colors) produced in two different locations (represented by different shapes). (B) Heatmap of relative abundance of Mf identified in the spermosphere clustered by Ward method. Each row represents a distinct metabolite feature and each column represents a distinct sample. The color intensity in each cell corresponds to the relative abundance of the respective Mf within the sample, with darker shades indicating higher abundance. (C) and (E) Bacterial and fungal community structures (β-diversity), respectively. Genotypes are represented by different colors and location of seed production by shapes. The variation in bacterial and fungal community structures explained by each PCoA axis is given in parentheses. (D) and (F) Heatmap of relative abundance of bacterial and fungal species in the spermosphere clustered by Ward method, respectively. Each row in the heatmap represents a distinct microbial species, while each column represents an individual sample. The color intensity in each cell corresponds to the relative abundance of the respective microbial species within the sample, with darker shades indicating higher abundance. (G) Permutational multivariate analysis of variance (PERMANOVA) showing the effects of genotype, location, and their interaction (G × L) on metabolome, bacterial, and fungal community structures. $R^2$ values represent the proportion of explained variance; asterisks denote significance ($P < 0.05$).

The top 15 nodes having the greatest degree (i.e., the number of connected nodes) are of microbial origin and are mainly represented by *Bacillus* spp. (Table 1).

## Complex interactions between microbial communities and metabolites within the germinating seed spermosphere

Specific correlation subnetworks for each major metabolic category (e.g., flavonoids) were constructed. More specifically, nodes of annotated metabolic categories correlated with microorganism nodes (first neighbors) were selected (sub-networks). The three largest correlation subnetworks belonged to amino acids and derivatives, flavonoids, and terpenoids and derivatives (Fig. 4B).

Enrichment analysis (hypergeometric test, $P < 0.05$, fold enrichment >1.5, Table 2) was carried out to identify over-represented bacterial and fungal genera for each annotated metabolic category sub-network (positive and negative correlations). *Bacillus* genus was over-represented within the subnetworks of amino acids and derivatives, flavonoids, terpenoids and derivatives, carbohydrates, and alkaloids. On the other hand, *Cladosporium* was over-represented across subnetworks of flavonoids, nucleosides and derivatives, cinnamic acids and derivatives, and terpenoids and derivatives sub-networks.

## Bacillus spp. are negatively correlated to different flavonoids

*Bacillus* was over-represented across multiple annotated metabolic categories and ranked among the most abundant genera in the common bean spermosphere. To further investigate its associations, a specific correlation subnetwork was constructed for *Bacillus* spp. (including all ASVs identified as *Bacillus*), capturing both positive and negative correlations. In total, 496 metabolite features (Mfs), of which 103 were annotated, showed positive correlations, while 91 Mfs (33 annotated) were negatively correlated with *Bacillus* spp. (see Table S10 at https://doi.org/10.6084/m9.figshare.26117287.v1). Flavonoids exhibited mainly negative correlations with *Bacillus* spp., whereas aa and peptides showed mainly positive correlations (Fig. 4C). *Bacillus* spp. were less abundant in the spermosphere of the CON and FAC genotypes (see Table S2 at https://doi.org/10.6084/m9.figshare.26117287.v1), which showed specific flavonoid accumulation profiles (Fig. 5A). Compared with the other genotypes, CON showed higher accumulation of flavan-3-ols, while FAC showed higher accumulation of anthocyanidins, flavonols, flavones, isoflavones, and aurones. Correlation analysis indicated that *Bacillus* spp. were negatively correlated with flavan-3-ols (including a putative catechin, p89) and anthocyanidins (including a putative malvidin-3-O-galactoside, p1858) (Fig. 4C and 5B). Catechin and malvidin-3-O-galactoside showed differential accumulation in different genotypes of common bean (Fig. 5B) and were negatively correlated with four and six *Bacillus* species, respectively (Fig. 5C). Experimental assay revealed a dose-dependent effect of catechin on *Bacillus megaterium* (B_ASV14) cultured in TSB with 10% DMSO. Indeed, at 0.1% catechin, growth rate increased; at 0.3% catechin, growth slowed down; and at 0.5% catechin, growth was completely inhibited (Fig. 5D).

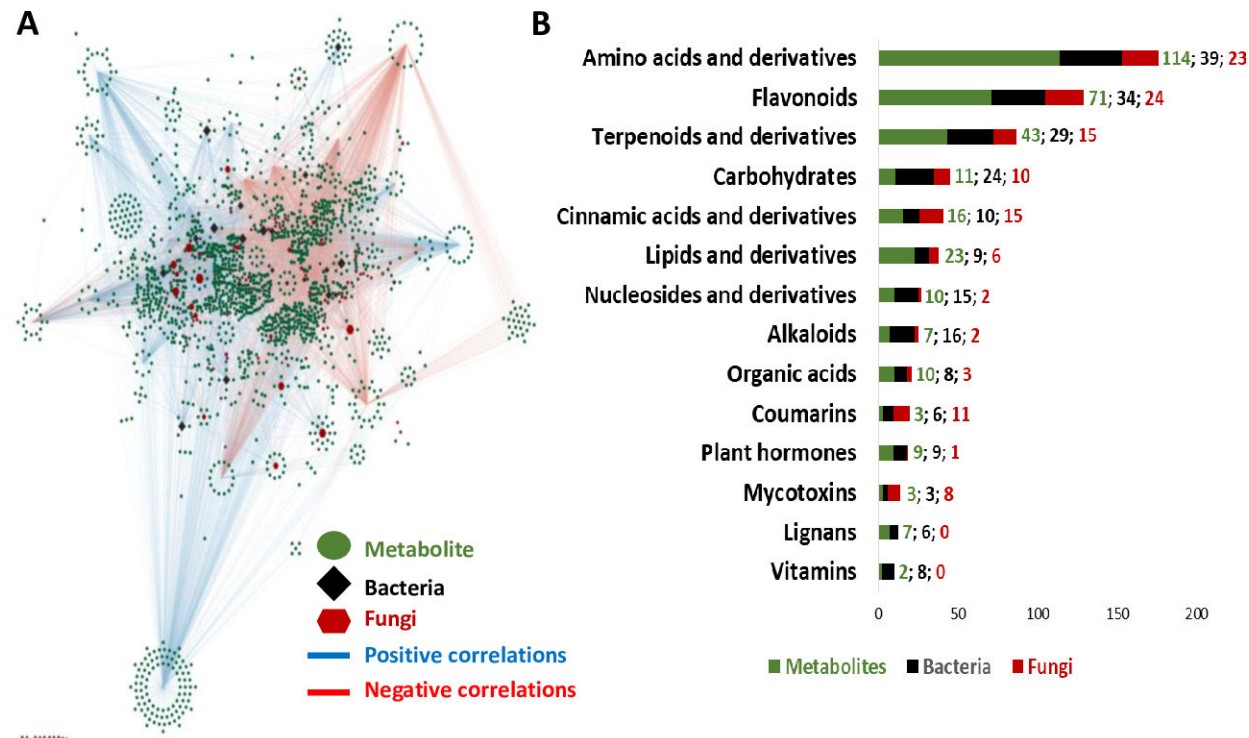

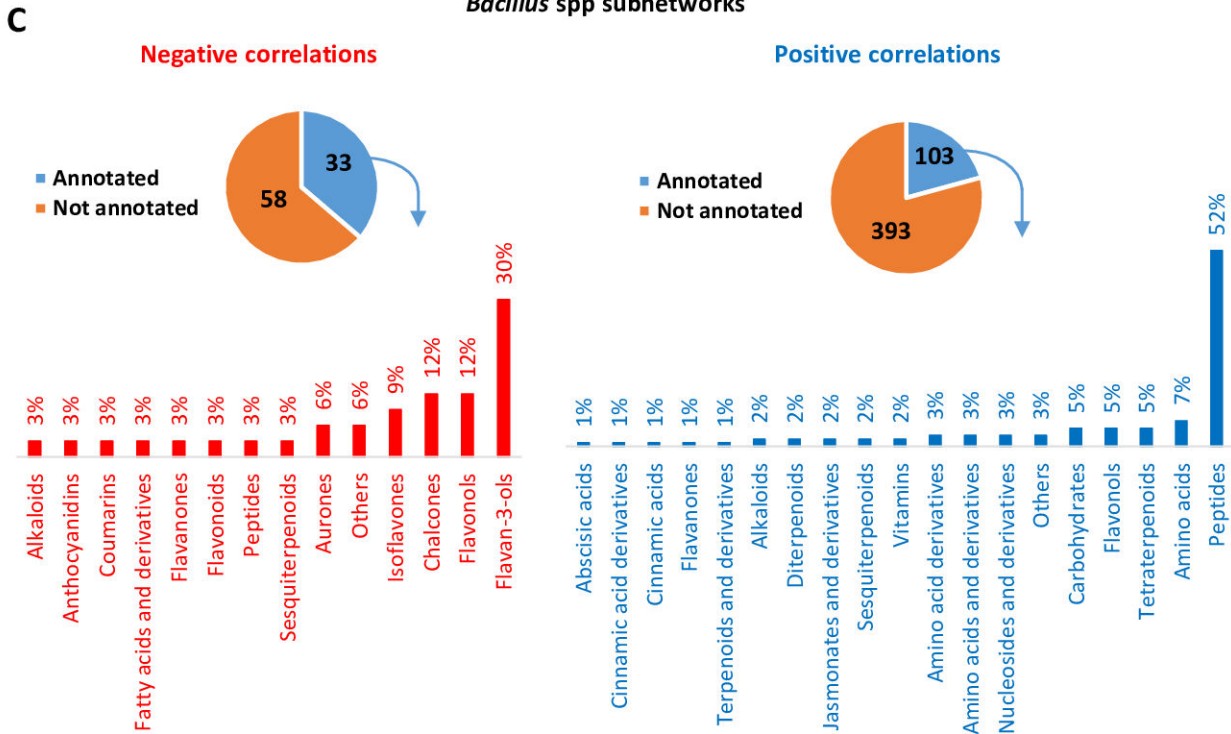

**FIG 4** Metabolite–microorganism correlation network analysis. (A) correlation network based on Spearman correlation (r > |0.5|, FDR *P*-value < 0.05) containing 1,736 nodes and 6,225 edges (see Table S10 at https://doi.org/10.6084/m9.figshare.26117287.v1). Only pairwise correlation microorganism-microorganism and metabolite-microorganism retained. Nodes in the network represent the metabolites (green circles), bacteria (black diamonds), and fungi (brown octagons). The size of each node in the network is proportional to their importance in terms of direct first neighbors they are connected to (degree). Positive correlations are represented by red edges, and inversely, blue edges represent negative correlations. (B) Size of metabolic category subnetwork in terms of number of total nodes (metabolites in green, bacteria in black, and fungi in brown). Metabolic subnetwork was extracted from the whole network and only microbial nodes

**Fig 4 (Continued)**

directly correlated to metabolic nodes (first neighbors) were selected. (C) *Bacillus* spp. subnetwork analysis revealed negative and positive correlations with annotated metabolites. Network based on Spearman correlation (r > |0.5|, *P* < 0.05). Positive and negative correlation networks are represented by orange and blue, respectively. *Bacillus* spp. positive correlation subnetwork consisted of 496 metabolites, of which 103 were assigned to a metabolic class. *Bacillus* spp. negative correlation subnetwork consisted of 91 metabolites, of which 33 were assigned to a metabolic class. Only the metabolites that have been assigned to a metabolic class are represented as a proportion in the graphic.

## DISCUSSION

The seed spermosphere acts as a crucial micro-environment surrounding seeds, displaying a specific microbial and chemical composition favorable for germination *sensu stricto* (5, 6). This environment may promote resilience against external biotic and abiotic stressors, facilitating the successful seedling establishment (5, 6, 29). To our knowledge, our study represents the first comprehensive characterization of both the chemical and microbial diversity of common bean spermosphere using an untargeted metabolomic and metabarcoding approaches.

Our work enabled the detection and characterization of an extensive range of metabolites, substantially enriching the limited existing data on the chemical diversity of the common bean spermosphere (16). Their exudation during germination raises questions about their functions. These metabolites were categorized into 14 major metabolic categories (see Table S1 at https://doi.org/10.6084/m9.figshare.26117287.v1), which were also characterized in dry common bean seeds (20). Primary and specialized metabolites are well-known compounds regulated by abiotic and biotic environments (30) and have been shown to influence microbiota in other plant compartments such as the rhizosphere when they are exuded (31, 32). These compounds included flavonoids (33), terpenoids (34), cinnamic acids (35), lipids (36), plant hormones (37), and alkaloids (38). Moreover, other compounds such as proteins (39, 40) and small peptides (5-50 aa) (24, 25) are also exuded by seeds during germination but not explored here. We hypothesize that these compounds play an important role in shaping the microbiome in the spermosphere during the early phases of germination.

As for the microbial community, the most abundant identified bacterial genera (i.e., *Pantoea, Leclercia, Bacillus, Paenibacillus, Escherichia, Pseudomonas*; see Table S8 at https://doi.org/10.6084/m9.figshare.26117287.v1) and fungal genera (i.e. *Alternaria, Stemphylium, Gibberella, Botrytis, Cladosporium, Cryptococcus, Monographella, Rhizoctonia*; see Table S9 at https://doi.org/10.6084/m9.figshare.26117287.v1) in the common bean spermosphere (Fig. 1C and G) have been previously identified in mature seeds (41), spermosphere, and young seedlings (17) of various plant species. These microorganisms may be potentially transmitted from dry seeds to the seedling, likely facilitated by their ability to establish and persist in the spermosphere environment (3, 29). Several abundant microbial genera identified in the spermosphere have been previously reported to promote seed germination and seedling growth (e.g., *Bacillus, Pantoea, Pseudomonas*) (42–44), while others are known to be associated with damping-off diseases in various plant species (e.g., *Alternaria, Botrytis, Stemphylium)* (45–48). However, the functional roles of microorganisms should be interpreted with caution, as both beneficial and pathogenic traits are context-dependent rather than inherent properties (49). Nonetheless, the high abundance of these genera may reflect their adaptation to the common bean spermosphere, potentially driven by a copiotrophic lifestyle and rapid growth in response to seed exudates (18, 50).

Our results highlighted a significant impact of genotype on both the chemical and microbial composition of the spermosphere (Fig. 3A, C and E). The high genetic variability in common bean, shaped by multiple domestication events and long-term genome evolution across diverse geographical regions (51, 52), may explain its strong genotype influence on chemical composition (20). In contrast, species with a shorter domestication and evolutionary history, such as rapeseed, show a metabolome more strongly influenced by the environment (53–55). This genetic diversity likely drives

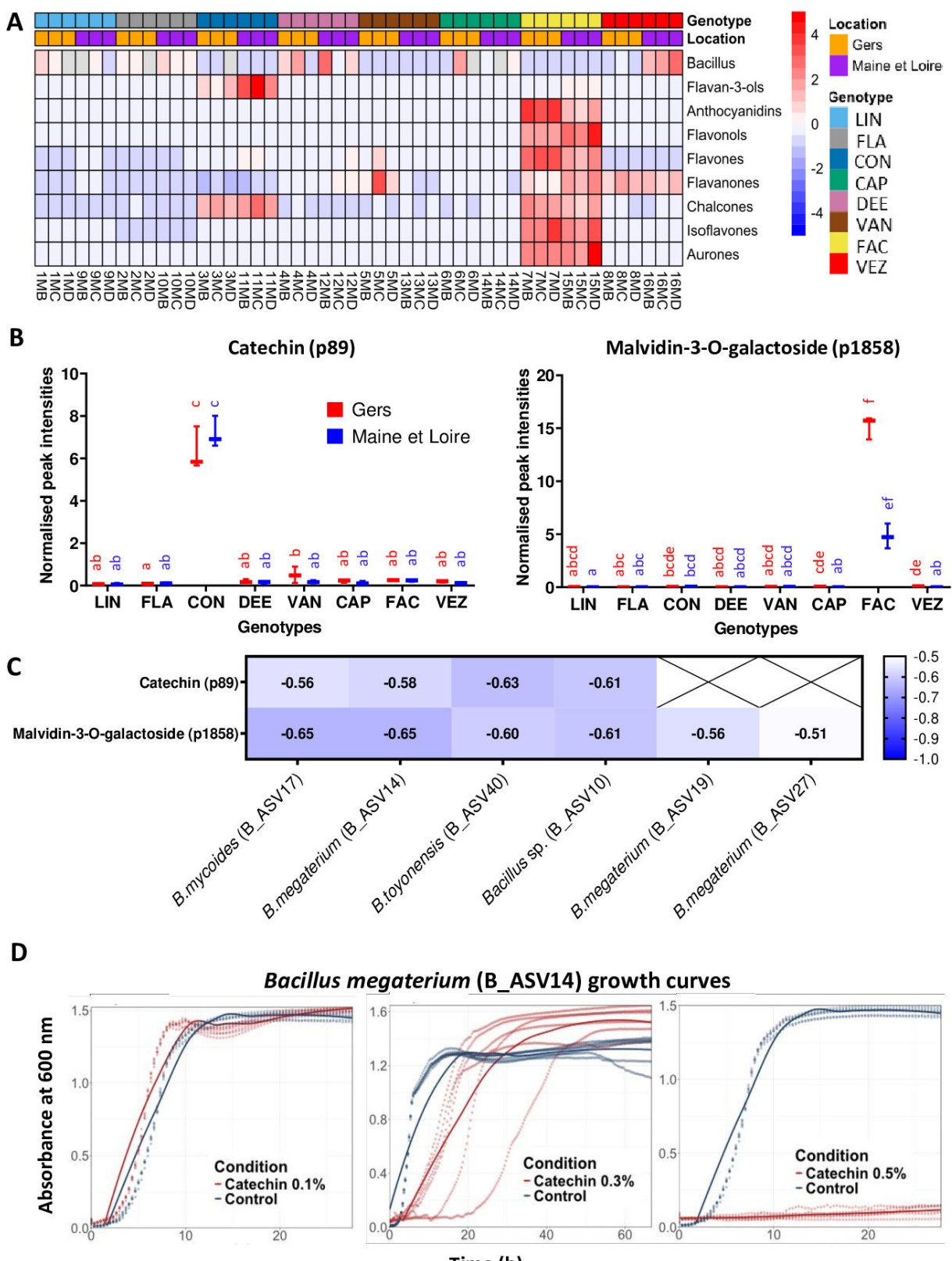

**FIG 5** Relationship between flavonoids and *Bacillus* spp. (A) Heatmap of relative abundance of *Bacillus* spp. and different flavonoid classes in the spermosphere of eight common bean genotypes produced in two locations. (B) Differential abundance of catechin (P89) and malvidin-3-O-galactoside (p1858) in the common bean spermosphere samples. Post-hoc Sidak test (*P* < 0.01) was carried out to assess differences in catechin and malvidin-3-O-galactoside composition among

Fig 5 (Continued)

genotypes and locations. (C) Correlation analysis heatmap between flavonoids (catechin and malvidin-3-O-galactoside) and *Bacillus* spp. Spearman correlation coefficient of each metabolite-microorganism pair shown on heatmap. Catechin and malvidin-3-O-galactoside are negatively correlated to four and six *Bacillus* species, respectively. (D) *In vitro* growth assay was conducted using the *Bacillus megaterium* strain (B_ASV14), isolated from common bean seed samples, to assess its response to catechin. Growth curves illustrate the response of the strain to commercial catechin at concentrations of 0.1%, 0.3%, and 0.5% (red), compared with a control condition with no catechin (blue). Each condition included six technical replicates, represented by dotted lines. The smooth solid curves correspond to LOESS-fitted average trends across these replicates for each condition. Absorbance was measured at 600 nm over time.

genotype-dependent variation in spermosphere chemistry, which, along with other factors, may contribute to the significant genotype effect on microbial composition, as previously observed in the rhizosphere (56–58). However, fungal communities were also and mainly affected by location of seed production in comparison to bacterial communities (Fig. 3C and E). This result could be explained by the relatively stable dynamics of fungal communities given their longer turnover times leading to fewer modifications of the community (59) and/or may be the possibility of an "endemism" effect of fungal communities based on the production area. In contrast, the genotype-dependent signature of the bacterial communities during germination could be a result of their faster multiplication and growth rate adapting to the seed exuded compounds in the spermosphere (60). Taken together, these results suggested a shift from predominantly environmental influences on dry seed microbial diversity (9) to a combined effect of genotype and environment in the spermosphere, possibly driven by seed exudate-mediated selection (3, 5, 6). This process may help conserve and transfer seed-associated microbial assemblages across generations (17, 61, 62).

Our multi-scale correlation network analysis revealed strong associations between *Bacillus* spp. and *Cladosporium* spp. with many central and specialized metabolites. Specifically, the negative correlation between flavonoids and *Bacillus* spp., known for their plant growth-promoting (PGP) bacteria properties (63), could be due to the composition of common bean seed coats. Specifically, CON and FAC genotypes, showing high accumulation of putative catechin (flavan-3-ol) and malvidin-galactoside (anthocyanidin), respectively, were the genotypes showing the lowest abundance of *Bacillus* subsp. (Fig. 5A) (see Fig. S1 at https://doi.org/10.6084/m9.figshare.26117287.v1). In this light, prior reports highlighted the antimicrobial properties of phenylpropanoids against *Bacillus* spp. (64). In addition, catechin has been shown to inhibit the growth of *B. subtilis* and *B. cereus,* while anthocyanin-rich extract demonstrated antimicrobial properties against *B. cereus* in a dose-dependent manner (65–67). Similarly, our experimental works confirmed a dose-dependent hormetic effect of catechin on *Bacillus megaterium* (B_ASV14), where low concentrations (0.1%) stimulated growth rate, while higher doses (0.5%) caused complete growth inhibition (Fig. 5D) as demonstrated with other flavonoids on different biological models (68). These findings reinforce the potential role of genotype-specific flavonoids in shaping *Bacillus* spp. abundance in the spermosphere. Although catechin was used as a representative flavonoid in our *in vitro* assay, it is likely that other structurally related compounds may exert similar effects on microbial growth. Future studies should evaluate a broader set of flavonoids to establish the generalizability of these effects on microbial community structure.

Due to their plant growth-promoting (PGP) properties, many studies have concentrated on using *Bacillus* spp. as seed coating or biopriming (69, 70). However, limited information is available about the factors contributing to their assembly in the spermosphere and promoting their transmission to the seedlings. Further research is needed to clarify the impact of these metabolites on *Bacillus* abundance, notably if flavonoids have a direct effect in shaping microbial communities during germination (5, 6), or indirect if these metabolites affected broader microbial communities, which could, in turn, impact *Bacillus* abundance through competitive interactions, niche exclusion, or microbial signaling (33). Additionally, the multi-scale correlation network analysis suggested a positive association between *Bacillus* and amino acids and small peptides,

potentially indicating their role as a carbon source (71). While these findings highlight key metabolic interactions, other correlations observed in the network warrant further investigation to uncover their ecological significance.

Our results support the original hypothesis that germinating seeds, through genotype-specific exudate profiles, play a central role in shaping the spermosphere by modulating both its metabolic composition and the assembly of associated microbial communities. While the approach used in this work does not fully replicate natural soil-microbe interactions in the spermosphere (3, 5, 6), the use of seed-dwelling microbes and water (24) ensured repeatability and enabled more precise metabolite character-ization. Additionally, by eliminating soil-related confounders, this method improved the detection of microbes present in seed exudates (72). Moreover, pooling exudates at different germination time points allowed capturing fluctuating metabolites and microbes and a more exhaustive characterization of seed exudates and microbes. Additionally, LC-MS/MS untargeted metabolomics and molecular networking improved metabolite annotation (73, 74) and, coupled to metabarcoding microbiome characteri-zation, generated an original data set that allowed the identification of many correla-tions between microorganisms and metabolites that need to be further explored (see Fig.S7 and S8; Table S10 at https://doi.org/10.6084/m9.figshare.26117287.v1). Further analysis of this novel data set and alternative multiscale strategies, or incorporating other network metrics like centrality measures, could provide deeper insights into these relationships (see Table S11 at https://doi.org/10.6084/m9.figshare.26117287.v1) (75–77). Overall, this study opens new avenues for further research into the metabolic capabilities and ecological roles of seed microbiota within the spermosphere to seed health and seed vigor.

## MATERIAL AND METHODS

### Biological materials

Seeds were harvested in the year 2020 from eight different common bean genotypes (French commercial cultivars of *Phaseolus vulgaris* L.) cultivated in two different locations from the FNAMS field experimental stations: Gers in South-West of France (43°57′25.2″N 0°23′31.7″E) and Maine et Loire (47°28′13.9″N 0°23′40.3″W) in the Loire Valley. The common bean genotypes, namely, VEZ, CON, CAP, FAC, FLA, LIN, DEE, and VAN were chosen to represent the highest diversity among breeding programs and qualitative seed traits in France by the seed company Vilmorin-Mikado SAS (Limagrain group). All seeds used in our experiment were from the same post-harvest age. The seeds underwent micro-cleaning, and a purity test was performed by GEVES (the French official seed testing organization) and eventually stored under identical conditions in accordance with the International Seed Testing Association (ISTA) recommendations (24). Seeds exhibit contrasting phenotypes in terms of size, weight, coat color and germina-tion kinetics as shown in Fig. S1 and S2 (see Fig. S1 and S2 at https://doi.org/10.6084/m9.figshare.26117287.v1).

### Spermosphere production and collection

The experimental design for collection of the spermosphere was as illustrated in Fig. S2 (see Fig. S2 at https://doi.org/10.6084/m9.figshare.26117287.v1) according to previously developed protocols (24) with modifications as detailed in Supplementary Methods 1.1 .

### Untargeted LC-MS/MS metabolomic analysis

Metabolite extraction was performed according to previously established protocols (78, 79). Similarly, data acquisition, processing, and annotations were also performed according to previously developed protocol with modifications as detailed in Supple-mentary Methods 1.2 (27).

## Metabarcoding and microbiome analysis

Microbial DNA extraction, library construction, data acquisition by sequencing, and data processing were adapted from a previous report with modifications as detailed in Supplementary Methods 1.3 (8).

## Statistical analysis

### Statistical analysis and computational analyses of polar and semi-polar specialized

All statistical analyses were conducted using R (v.4.1.1). The metabolomic data set, consisting of 48 samples, was normalized and then log10-transformed. A linear model was fitted to the log-transformed data to assess the effects of genotype, location, and their interaction (G × L) on metabolite accumulation. Two-way analysis of variance (ANOVA) was performed to evaluate for significant effects of genotype, location, and their interaction (G × L). To address multiple testing, the $P$-values from the ANOVA were adjusted using the false discovery rate (FDR) correction according to the Benjamini-Hochberg method. This controlled for Type I errors across multiple tests. The adjusted $P$-values for the interaction, genotype, and location effects were reported in the final analysis.

Principal component analysis (PCA) was conducted on the normalized metabolomic data set comprising 48 samples. The results were visualized in PCA plots, where different genotypes were color-coded, and locations were represented by different shapes. Heatmaps and hierarchical clustering analyzes of metabolomic data were carried out using the pheatmap R package (v.1.0.12) (https://cran.r-project.org/package=pheatmap) (80). The heatmaps were built using the following parameters: Euclidean distance measure and Ward clustering method.

### Microbial community analysis

Prior to statistical analysis, rarefaction was performed on the metabarcoding data, resulting in 42 bacterial samples (with six samples lost) and 48 fungal samples (see Fig. S3 at https://doi.org/10.6084/m9.figshare.26117287.v1). Analyses of microbial diversity were then carried out with the R package "phyloseq" (v.1.38.0) (81). Microbial richness was assessed using the number of amplicon sequence variants (ASVs) per sample. Analysis of variance (ANOVA) was performed to assess the significant influences of genotype, location, and their interaction on microbial richness. To further examine the significant differences in microbial richness attributed to genotype within each location, post-hoc tests were conducted using Tukey's honestly significant difference (HSD) method. Changes in community composition were assessed using Bray-Curtis dissimilarity. The relative contributions of plant genotypes and production region in community composition were estimated with permutational multivariate analysis of variance (PERMANOVA) (82) through the function adonis2 of the R package vegan v.2.6.4 (83). All figures were prepared using ggplot2 v.3.4.3 (84), and the data management was done using dplyr v.1.1.3 and tidyverse v.2.0.0. Heatmaps were generated using the plot_taxa_heatmap function available in the package microbiomeutilities v.1.00.17 (85) from the microbial abundance count data.

## Integration of metabolome and microbiome analyses

Since both metabolomic and metabarcoding analyses were performed on the same biological samples, we applied a correlation-based multiscale integration approach to explore potential associations between metabolites and microbial communities in the spermosphere (28, 86). While correlation analysis can reveal significant associations, it does not establish causation (87). However, multi-omics integration is a powerful tool for generating hypotheses about potential biological interactions and underlying mechanisms (88).

## Correlation-based approaches

To integrate the metabolomic and microbiome results, the two normalized data sets were combined and standardized by scaling, which involved centering each variable to have a mean of 0 and scaling them to have a standard deviation of 1. This step removed the influence of scale differences among variables, enabling meaningful comparisons and analyzes. Finally, Spearman correlation coefficients were computed on the standardized data set to assess relationships between variables while accounting for potential non-linear associations. All $P$-values in the Spearman correlation were adjusted by Benjamini and Hochberg using the false discovery rate (FDR) control procedure. We focused on microbial-microbial and microorganism-metabolite correlations ($P < 0.05$) to investigate community dynamics and the influence of seed-exuded metabolites on microbial assembly, while metabolite-metabolite correlations were omitted as they were not the primary focus of our analysis (see Table S10 at https://doi.org/10.6084/m9.figshare.26117287.v1). Consequently, only microbial-microbial and microorganism-metabolite pairs were retained if their correlation coefficient was ≥ ⎹ 0.5 ⎹, which were considered moderate to strong associations (see Table S10 at https://doi.org/10.6084/m9.figshare.26117287.v1) (89, 90).

## Network analysis and enrichment analysis

A pairwise correlation network was visualized from microbial-microbial and microorganism-metabolite pairs of correlation coefficient ≥ ⎹ 0.5 ⎹ using Cytoscape (v.3.10.0) (91). Metabolic subnetworks were extracted from the previously created pairwise correlation network comprising both positive and negative correlations. From the initial network, annotated metabolic nodes of each major metabolic category were selected along with their direct correlated microorganism node (first neighbors). Enrichment analysis using hypergeometric test was carried out by comparing microbial genera in the different metabolic category correlation subnetworks to those present in the spermosphere. For enrichment analysis, reference sets were bacterial and fungal ASVs identified in spermosphere respectively, and the target sets were bacterial and fungal ASVs in the respective metabolic categories' subnetworks. Network analysis was also carried out using the Cytoscape built-in plugin Network Analyzer to calculate node degree, a key metric representing the number of connections (edges) each node has within the network (91). Node degree was chosen as it directly quantifies interaction frequency, allowing us to identify highly connected microbial taxa and metabolites that may play key roles in spermosphere dynamics. Additional network metrics were also calculated with Cytoscape and are provided in the supplementary material, although they were not specifically explored in this study (see Table S11 at https://doi.org/10.6084/m9.figshare.26117287.v1).

## *In vitro* experimental investigation of the effects of catechin on *Bacillus megaterium*

*Bacillus megaterium* strain B_ASV14 was isolated on Tryptic Soy Agar (TSA) 10% strength from the same common bean seed lot used in the study (CON produced in Maine et Loire) and registered in the CIRM-CFBP French Collection for Plant-Associated Bacteria under the accession numbers CFBP9575. The strain was first grown on Tryptic Soy Broth (TSB) agar plates (Fisher BioReagents BP2471-100) and incubated at 28°C for 24 h. A single colony was then inoculated into 10 mL of liquid TSB medium and incubated under agitation. After 12 h, the optical density ($OD_{600}$) of the culture was measured using an Eppendorf Biophotometer Plus with square cuvettes. Cultures with an $OD_{600}$ between 0.4 and 0.8 were diluted to an $OD_{600}$ of 0.01, then transferred to a 96-well microplate for growth assays. Experimental conditions included a control (TSB containing 10% DMSO, CAS 67-68-5, without catechin), and catechin treatments (CAS 225937-10-0) at final concentrations of 0.1%, 0.3%, and 0.5% (w/v), each prepared in TSB containing 10% DMSO to ensure identical solvent conditions across treatments. Each condition was

tested in six technical replicates. The microplate was incubated at 28°C for 72 h with continuous shaking, and absorbance at 600 nm was recorded every 20 min using a TECAN Spark spectrophotometer (92, 93).

## ACKNOWLEDGMENTS

This work was also part of the 3rd Program for Future Investments (France2030) and operated by the SUCSEED project (ANR-20-PCPA-0009) funded by the 'Growing and Protecting crops Differently' French Priority Research Program (PPR-CPA), part of the national investment plan operated by the French National Research Agency (ANR). This work has benefited from the support of IJPB's Plant Observatory platform PO-Chem (Metabolomic platform) and PO-Bioch (biochemistry platform). The IJPB benefits from the support of Saclay Plant Sciences-SPS (ANR-17-EUR-0007).

## AUTHOR AFFILIATIONS

[1]Université Paris-Saclay, INRAE, AgroParisTech, Institute Jean-Pierre Bourgin for Plant Sciences (IJPB), Versailles, France
[2]IRHS-UMR1345, Université d'Angers, INRAE, Institut Agro, Beaucouzé, France

## AUTHOR ORCIDs

Chandrodhay Saccaram https://orcid.org/0009-0000-9366-6386
Marie Simonin http://orcid.org/0000-0003-1493-881X
Matthieu Barret http://orcid.org/0000-0002-7633-8476
Loïc Rajjou http://orcid.org/0000-0001-9739-1041
Massimiliano Corso http://orcid.org/0000-0002-3243-1660

## FUNDING

| Funder | Grant(s) | Author(s) |
| --- | --- | --- |
| Agence Nationale de la Recherche | ANR-20-PCPA-0009 | Chandrodhay Saccaram |
| | | Marie Simonin |
| | | Matthieu Barret |
| | | Loïc Rajjou |
| Agence Nationale de la Recherche | ANR-17-EUR-0007 | Chandrodhay Saccaram |
| | | Loïc Rajjou |

## AUTHOR CONTRIBUTIONS

Chandrodhay Saccaram, Data curation, Formal analysis, Investigation, Methodology, Validation, Visualization, Writing – original draft | Marie Simonin, Conceptualization, Data curation, Funding acquisition, Investigation, Methodology, Project administration, Resources, Supervision, Validation, Visualization, Writing – review and editing | Stéphanie Boutet, Data curation, Formal analysis, Investigation, Methodology, Resources, Supervision, Validation, Visualization, Writing – review and editing | Céline Brosse, Data curation, Formal analysis, Investigation, Resources, Visualization | Shuang Peng, Data curation, Formal analysis | Tracy François, Formal analysis, Investigation, Methodology, Supervision | Boris Collet, Formal analysis, Investigation, Methodology, Supervision | François Perreau, Formal analysis, Investigation, Methodology, Resources, Writing – original draft | Delphine Sourdeval, Formal analysis, Investigation, Methodology | Coralie Marais, Formal analysis, Investigation, Resources | Matthieu Barret, Funding acquisition, Project administration, Resources, Validation, Writing – review and editing | Loïc Rajjou, Conceptualization, Formal analysis, Funding acquisition, Investigation, Methodology, Project administration, Resources, Supervision, Validation, Visualization, Writing – original draft, Writing – review and editing | Massimiliano Corso, Data curation, Formal

analysis, Investigation, Resources, Supervision, Validation, Visualization, Writing – original draft, Writing – review and editing

## DATA AVAILABILITY

The data that support the findings of this study are openly available in online repositories. Metabolomic LC-MS/MS data are available on MassIVE at https://massive.ucsd.edu/, reference number MSV000094600 and the metaboarcoding data are available in the European Nucleotide Archive (ENA) at https://www.ebi.ac.uk/ena/, reference number PRJEB59579. The supplemental material is available on Figshare at https://doi.org/10.6084/m9.figshare.26117287.v1.

## ADDITIONAL FILES

The following material is available online.

### Open Peer Review

**PEER REVIEW HISTORY (review-history.pdf).** An accounting of the reviewer comments and feedback.

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
