## [Reviewer comments · mSystems]

Elucidating the interplay between metabolites and microorganisms in the spermosphere of common bean (*Phaseolus vulgaris* L.) seeds

Chandrodhay Saccaram, Marie Simonin, Stéphanie Boutet, Céline Brosse, Shuang Peng, Tracy François, Boris Collet, François Perreau, Delphine Sourdeval, Coralie Marais, Matthieu Barret, Loic Rajjou, and Massimiliano Corso

Corresponding Author(s): Loic Rajjou, Institut Jean-Pierre Bourgin

Review Timeline:

Submission Date:	May 19, 2025
Editorial Decision:	June 12, 2025
Revision Received:	June 22, 2025
Accepted:	June 26, 2025

Editor: Christopher Anderton

Reviewer(s): Disclosure of reviewer identity is with reference to reviewer comments included in decision letter(s). The following individuals involved in review of your submission have agreed to reveal their identity: Malak Tfaily (Reviewer #1); Toshihiro Obata (Reviewer #2)

Transaction Report:

DOI: <https://doi.org/10.1128/msystems.00707-25>

Re: mSystems00707-25 (**Elucidating the interplay between metabolites and microorganisms in the spermosphere of common bean (*Phaseolus vulgaris* L.) seeds**)

Dear Prof. Loic Rajjou:

Revision Guidelines

Sincerely,
Christopher Anderton
Editor
mSystems

Reviewer #1 (Comments for the Author):

I would like to thank the authors for their detailed and thoughtful response to my comments, as well as for the extensive revisions made to the manuscript. In particular, I was pleased to see the inclusion of the full protocol for metabolite annotation. Considering that the authors have addressed my comments as requested and that their responses were clear and satisfactory, I have no further comments.

Reviewer #2 (Comments for the Author):

The revised manuscript addressed my most critical concerns in the original manuscript. Now the discussion is not overstated, and a newly presented experiment showed the effect of catechin on the growth of *B. begaterium* species at high concentrations. However, to support the catechin's effects on the microbial community compositions, test with more than one metabolites is ideal because catechin likely decreases growth of other bacterial species too, as stated by the authors (L270).

Minor concerns

L32: "Thousands of metabolites" is unspecific. Specify the number. Common bean exudate very unlikely contains thousands of metabolites. They should be described as metabolic features or chromatography peaks.

L88-90: The hypothesis should be properly addressed in the discussion.

L215: Broader than what?

L265: "inhibit *B. subtilis* and *B.cereus*" is strange.

L267 and others: Consider the use of "experimental validation". Validation of what?

L387, L737: The description of the growth experiment is awkward. The DMSO concentrations of the control and catechin treatments do not match.

Massimiliano CORSO, PhD, Researcher
Loïc RAJJOU, PhD, Professor
Institute Jean-Pierre Bourgin for Plant
Sciences (IJPB)
INRAE- Versailles
FRANCE
massimiliano.corso@inrae.fr
loic.rajjou@agroparistech.fr

June 19th, 2025

To mSystems Editorial Board

Response to editors & reviewers, Saccaram and colleagues (mSystems00707-25)

Dear Editor,

Thank you for reviewing our manuscript entitled "Elucidating the interplay between metabolites and microorganisms in the spermosphere of common bean (*Phaseolus vulgaris* L.) seeds" (revised version **mSystems00707-25** ; initial submission mSystems00870-24).

We have addressed the reviewers' remarks thanks to which the manuscript quality has been improved. In the revised manuscript, we have included Delphine Sourdeval as a co-author, in recognition of her significant contributions to the additional experiments and data interpretation introduced in response to previous reviewer feedback. We have included a copy of the manuscript with track changes.

Point-by-point response to the reviewers

Reviewer #1:

Comments for the Author: I would like to thank the authors for their detailed and thoughtful response to my comments, as well as for the extensive revisions made to the manuscript. In particular, I was pleased to see the inclusion of the full protocol for metabolite annotation. Considering that the authors have addressed my comments as requested and that their responses were clear and satisfactory, I have no further comments.

Authors' response: We would like to express our sincere gratitude to Reviewer 1 for the high quality, rigour, and scientific relevance of their evaluation. Their careful reading of our work, along with the constructive and detailed comments provided, have greatly contributed to improving the manuscript, both in terms of content and clarity. Thanks to their insightful suggestions, we were able to clarify key methodological aspects, enhance the presentation of our results, and strengthen the overall robustness of the study. We sincerely hope that these revisions will render the manuscript suitable for publication in mSystems.

Reviewer #2:

Comments for the Author: The revised manuscript addressed my most critical concerns in the original manuscript. Now the discussion is not overstated, and a newly presented experiment showed the effect of catechin on the growth of *B. megaterium* species at high concentrations. However, to support the catechin's effects on the microbial community compositions, testing more than one metabolite is ideal because catechin likely decreases growth of other bacterial species too, as stated by the authors (L270).

Authors' Response: We sincerely appreciate this constructive feedback and thoughtful suggestions on our revised manuscript. Below, we address the comments point by point. All changes have been incorporated into the revised version, with updated line numbers as indicated. We agree that extending the analysis to additional metabolites would strengthen the findings. While we focused on catechin as a proof-of-concept due to its strong negative correlations with *Bacillus* spp., we have now explicitly acknowledged this limitation in the Discussion and emphasized the need for future testing of additional flavonoids to support broader applicability.

Revision added to the manuscript (Discussion, lines 271–274): “Although catechin was used as a representative flavonoid in our *in vitro* assay, it is likely that other structurally related compounds may exert similar effects on microbial growth. Future studies should evaluate a broader set of flavonoids to establish the generalizability of these effects on microbial community structure.”

Minor Concerns :

Reviewer #2 - L32: "Thousands of metabolites" is unspecific. Specify the number. Common bean exudate very unlikely contains thousands of metabolites. They should be described as metabolic features or chromatography peaks.

Authors' Response: Thank you for this correction. We revised the sentence to reflect the accurate terminology and to specify the number detected.

Revised sentence (L32): “We detected and analysed 2,467 metabolite features (Mf) through untargeted metabolomics...”

Reviewer #2 - L88–90: The hypothesis should be properly addressed in the discussion.

Authors' Response: While the Discussion interprets the data in light of genotype and seed-mediated influences via exudated metabolites, we have now explicitly linked our findings back to the hypothesis stated in the Introduction.

Revision added to the Discussion (286-288): “Our results support the original hypothesis that germinating seeds, through genotype-specific exudate profiles, play a central role in shaping the spermosphere by modulating both its metabolic composition and the assembly of associated microbial communities.”

Reviewer #2 - L215: "Broader than what?"

Authors' Response: We appreciate the reviewer pointing out this ambiguity. We revised the sentence to clarify that the “broader range” refers to an expansion beyond previous studies.

Revised sentence (L215–216): “Our work enabled the detection and characterization of an extensive range of metabolites, substantially enriching the limited existing data on the chemical diversity of the common bean spermosphere.”

Reviewer #2 - L265: "inhibit B. subtilis and B.cereus" is strange.

Authors' Response: We revised the phrasing to be more precise.

Revised sentence (L265): “In addition, catechin has been shown to inhibit the growth of *B. subtilis* and *B. cereus*...”

**Reviewer #2 - L267 and others: Consider the use of "experimental validation".
Validation of what?**

Authors' Response: It is correct that “validation” implies a confirmatory study. We have replaced “experimental validation” with alternative wording to better reflect the exploratory nature of the catechin growth assay.

- **Revised sentence (L203):** « Experimental assay... »
- **Revised sentence (L267):** « our experimental work... »
- **Revised sentence (L392):** « *In-vitro* experimental investigation of the effects of catechin on *Bacillus megaterium* »
- **Revised sentence (L743-744):** « *In vitro* growth assay was conducted using the *Bacillus megaterium* »

Reviewer #2 - L387, L737: The description of the growth experiment is awkward. The DMSO concentrations of the control and catechin treatments do not match.

Authors' Response:

Thank you for catching this. We confirm that all catechin treatments were dissolved in 10% DMSO, and the control also used 10% DMSO to ensure identical solvent conditions across treatments. We clarified the sentence accordingly.

Revised sentence (L400–403):

“Experimental conditions included a control (TSB containing 10% DMSO, CAS 67-68-5, without catechin), and catechin treatments (CAS 225937-10-0) at final concentrations of 0.1%, 0.3%, and 0.5% (w/v), each prepared in TSB containing 10% DMSO to ensure identical solvent conditions across treatments.”

Re: mSystems00707-25R1 (**Elucidating the interplay between metabolites and microorganisms in the spermosphere of common bean (*Phaseolus vulgaris* L.) seeds**)

Dear Prof. Loic Rajjou:

Your manuscript has been accepted, and I am forwarding it to the ASM production staff for publication. Your paper will first be checked to make sure all elements meet the technical requirements. ASM staff will contact you if anything needs to be revised before copyediting and production can begin. Otherwise, you will be notified when your proofs are ready to be viewed.

Sincerely,
Christopher Anderton
Editor
mSystems